# Comparative Mitogenome Analyses of Fifteen Ramshorn Snails and Insights into the Phylogeny of *Planorbidae* (Gastropoda: Hygrophila)

**DOI:** 10.3390/ijms25042279

**Published:** 2024-02-14

**Authors:** Kexin Tao, Yue Gao, Haofei Yin, Qichao Liang, Qianqian Yang, Xiaoping Yu

**Affiliations:** Zhejiang Provincial Key Laboratory of Biometrology and Inspection & Quarantine, College of Life Sciences, China Jiliang University, Hangzhou 310018, China; s21090710048@cjlu.edu.cn (K.T.); s21090710013@cjlu.edu.cn (Y.G.); s22090710066@cjlu.edu.cn (H.Y.); s23090710022@cjlu.edu.cn (Q.L.)

**Keywords:** ramshorn snails, mitochondrial genes, taxonomy, phylogenetic relationships, gene rearrangement

## Abstract

Ramshorn snails from the family *Planorbidae* are important freshwater snails due to their low trophic level, and some of them act as intermediate hosts for zoonotic trematodes. There are about 250 species from 40 genera of *Planorbidae*, but only 14 species from 5 genera (*Anisus*, *Biomphalaria*, *Bulinus*, *Gyraulus*, and *Planorbella*) have sequenced complete mitochondrial genomes (mitogenomes). In this study, we sequenced and assembled a high-quality mitogenome of a ramshorn snail, *Polypylis* sp. TS-2018, which represented the first mitogenome of the genus. The mitogenome of *Polypylis* sp. TS-2018 is 13,749 bp in length, which is shorter than that of most gastropods. It contains 13 protein-coding genes (PCGs), 22 transfer RNA (tRNA) genes, and 2 ribosomal RNA (rRNA). We compared mitogenome characteristics, selection pressure, and gene rearrangement among all of the available mitogenomes of ramshorn snails. We found that the nonsynonymous and synonymous substitution rates (Ka/Ks) of most PCGs indicated purifying and negative selection, except for *atp8* of *Anisus*, *Biomphalaria*, and *Gyraulus*, which indicated positive selection. We observed that transpositions and reverse transpositions occurred on 10 tRNAs and *rrnS*, which resulted in six gene arrangement types. We reconstructed the phylogenetic trees using the sequences of PCGs and rRNAs and strongly supported the monophyly of each genus, as well as three tribes in *Planorbidae*. Both the gene rearrangement and phylogenetic results suggested that *Polypylis* had a close relationship with *Anisus* and *Gyraulus*, while *Bulinus* was the sister group to all of the other genera. Our results provide useful data for further investigation of species identification, population genetics, and phylogenetics among ramshorn snails.

## 1. Introduction

Studying molecular variation and phylogenetics among closely related species will provide insights into basic biological questions, such as those pertaining to taxonomy [1], genetic diversity [2], evolution [3], and interspecific interaction relationships [4]. With the development of high-throughput sequencing and bioinformatic analyses, the bulk of molecular data has become efficient and economical to obtain, which allows them to be applied to the resolution of the longstanding questions posed by traditional techniques [5]. For example, traditional snail species identification in mollusks has largely depended on shell morphology combined with limited anatomical characteristics (e.g., male and female reproductive system of adults), which has resulted in confusion due to geographically varying intraspecific morphology [6,7].

The ramshorn snails from the family *Planorbidae* (Gastropoda: Hygrophila), a group of minute air-breathing freshwater species, are widely distributed in various freshwater ecosystems around the world, including ponds, lakes, rivers, and streams [8,9]. The family contains about 250 species from 40 genera [9,10]. The herbivorous ramshorn snails are food sources for predators, significantly contributing to the overall biodiversity and stability of food webs in freshwater ecosystems [9] Some *Planorbidae* species are introduced outside their native ranges and become invasive, causing serious damage to natural aquatic ecosystems [11,12]. In addition, the most crucial practical importance of many *Planorbidae* species is related to their parasitological impacts. For example, the ramshorn snails from the genus *Biomphalaria* and *Bulinus* act as the obligate intermediate host of the blood flukes *Schistosoma mansoni* [13] and *S. haematobium* [14], respectively. This parasitic relationship enables the transmission of schistosomes to humans and domestic and wild animals, which causes a chronic debilitating disease known as schistosomiasis and poses a serious threat to public health [13,14]. The *Polypylis* species are also intermediate hosts for several zoonotic trematodes, notably the human intestinal fluke *Neodiplostomum seoulensis* [15].

Due to the high economic and health importance of the ramshorn snails, a couple of studies on their taxonomy and phylogenetic relationships have largely resolved confusion regarding the family *Planorbidae*, especially at higher taxonomic levels. Baker [8] divided *Planorbidae* into 36 genera of four subfamilies (Planorbinae, Segmentininae, Helisomatinae, and Planorbulinae) based on conchological and anatomical characters. Hubendick [16] emphasized the phylogenetic implications of male reproductive morphology and divided *Planorbidae* into 31 genera of three subfamilies: Bulininae + Planorbinae formed the most apical clade, while Plesiophysinae occupied the basal positions. Hubendick [17] suggested combining the diverse species from the families Ancylidae and *Planorbidae* to form the gigantic family *Planorbidae* and rearranged the subfamilies into Ancylinae, Bulininae, and Planorbinae, which was confirmed by the recent molecular phylogeny using the genes cytochrome *c* oxidase subunit I (*cox1*), large subunit ribosomal RNA (*rrnL*), *18S rDNA*, *28S rDNA*, and actin exon 2 [10,18,19]. However, Bouchet and Rocroi [20] combined previous studies and provided a classification scheme of *Planorbidae*, in which they divided *Planorbidae* into three subfamilies: Ancylinae, Miratestinae, and Planorbinae. Planorbinae was the most diverse subfamily, consisting of Camptoceratini, Coretini, Drepanotrematini, Helisomatini, Neoplanorbini, Planorbini, and Segmentinini. In recent years, a couple of molecular phylogenetic studies of *Planorbidae* have been reported, but mainly focused on two economically important genera, i.e., *Biomphalaria* and *Bulinus*, that infer the phylogeographic relationship using the mitochondrial gene of *cox1* and *rrnL*, and nuclear gene *18S rDNA*, *28 S rDNA* and internal transcribed spacer (ITS) [21,22,23].

The mitochondrial genome (mitogenome) of gastropods is a DNA molecule that is usually 13–20 kb in size, with 37 genes, including 13 protein-coding genes (PCGs), 22 transfer RNA genes (tRNAs), 2 ribosomal RNA genes (rRNAs) and usually longer noncoding regions (control regions) [24]. Due to maternal inheritance, a high mutation rate, and a lack of recombination, the mitogenome is widely considered one of the most reliable and effective genetic markers in species delimitation, demographic and phylogenetic history analyses, introgression studies, and phylogeographical investigations [3,25,26]. In recent years, 32 complete mitogenome sequences of 14 species from five genera of *Planorbidae* have been published in GenBank (http://www.ncbi.nlm.nih.gov/, accessed on 15 October 2023) [27]. Meanwhile, there has been no report on the phylogenetic relationships of *Planorbidae* based on the complete mitogenome datasets at the family level.

In this study, we aimed to (1) enrich the mitogenomic data of *Planorbidae*; (2) evaluate the variation and conservation of mitogenomes of ramshorn snails; and (3) further investigate phylogenetic relationships among *Planorbidae* species. We reported the complete mitogenome sequence of *Polypylis* sp. TS-2018 for the first time and compared it with all the published mitogenomes of *Planorbidae* species. Moreover, we reconstructed the phylogenetic trees of all of the available mitogenomes from *Planorbidae* to provide a new perspective from which to discuss their relationships.

## 2. Results and Discussion

### 2.1. Species Identification

The *cox1* gene has been widely used as a DNA barcode for species identification in animals [28]. We sequenced and obtained a *cox1* barcoding region of 641 bp, which showed a 100% sequence similarity to the *cox1* gene sequence of *Polypylis* sp. TS-2018 (GenBank accession number LC429530) published by Saito et al. [7]. Saito et al. [7] collected 205 ramshorn snails from 163 sites in Japan, revealing that *Polypylis* sp. TS-2018 formed a monophyletic clade with other *Polypylis* species on the phylogenetic trees by using *cox1*, *rrnL*, and the nuclear histone 3 (H3). Thus, we confirmed the ramshorn snail specimen used in this study as *Polypylis* sp. TS-2018. Our study might indicate that this undescribed *Polypylis* sp. TS-2018 can be found in China in addition to Japan. *Polypylis* has a wide species diversity in Eastern Asia, with only a couple of species being described and many invalid species listed [7]. Due to the incomplete taxonomy and limited molecular data available regarding *Polypylis*, we were limited to identifying the species at the genus level. It is possible that *Polypylis* sp. TS-2018 represents a described species without a DNA barcode, and the investigation is beyond the scope of our research. Therefore, further investigation is needed to clarify the taxonomic status of *Polypylis* species with integrative morphological and molecular studies.

### 2.2. Mitogenome Structure and Organization

The complete mitogenome of *Polypylis* sp. TS-2018 is 13,746 bp in length (GenBank accession number OR684570), which is within the range of the reported mitogenomes in *Planorbidae*, varying between 13,566 bp in *Anisus vortex* [29] and 13,747 bp in *Bulinus globosus* [14]. The arrangement of genes in the *Polypylis* sp. TS-2018 mitogenome is compact. There are 14 intergenic spacers and 10 overlap regions in the whole mitogenome with total lengths of 107 bp and 68 bp, respectively. The intergenic regions occur at 14 gene junctions with the longest intergenic spacer between *cox3* and *trnI* (43 bp). The overlap regions occur at 12 gene junctions with the longest overlap region between *nad1* and *nad5* (20 bp) (Table 1). The longest intergenic spacer has also been reported in other *Planorbidae* species, such as *Biomphalaria glabrata* [13], *B. pfeifferi* [13], *B. straminea* [13], *Bulinus ugandae* [14], and *Planorbella pilsbryil* [30]; while the longest overlap region has been observed in the mitogenome of *Anisus vortex* [29], with a similar length to *Polypylis* sp. TS-2018 (Appendix A). Interestingly, we observe that the mitogenomes of ramshorn snails are shorter than all of other known gastropods except that of *Melibe japonica*, which is 13,216 bp in size [31]. The compact mitogenome structure, characterized by long overlap regions and small intergenic spacers between adjacent genes, may contribute to the relatively small size of ramshorn snail mitogenomes within the gastropod group. In addition, most of the size variation among animal mitogenomes is due to differences in the number of noncoding regions [32]. Bilaterian animals possess a large noncoding region referred to as the “control region”, or “D-loop” in mammals [32]. Non-bilaterian animals distribute noncoding nucleotides more evenly among the intergenic regions [32]. The absence of an identified control region may contribute to the small size of ramshorn snails.

The mitogenome of *Polypylis* sp. TS-2018 exhibits strong AT bias with a base composition of 33.3% A, 42.0% T, 11.5% C, and 13.3% G. The nucleotide skew statistics show a negative AT skew (−0.116) and a positive GC skew (0.075), indicating an obvious bias toward the use of Ts and a slight bias toward Gs in the whole mitogenome. The mitogenome of *Polypylis* sp. TS-2018 encodes 13 PCGs, 22 tRNAs, and 2 rRNAs. Among these genes, 25 genes located on the majority strand (J-strand) including 9 PCGs (*cob*, *cox1*, *cox2*, *nad1*, *nad2*, *nad4*, *nad4L*, *nad5*, *nad6*), 14 tRNAs (*trnA*, *trnC*, *trnD*, *trnF*, *trnG*, *trnH*, *trnI*, *trnK*, *trnL1*, *trnP*, *trnS1*, *trnV*, *trnW*, *trnY*), and 2 rRNAs (*rrnS* and *rrnL*); while the other genes are encoded by the minority strand (N-strand) (Figure 1). Our data suggest that the mitogenome characteristics of *Polypylis* sp. TS-2018 are conserved in other reported ramshorn snails.

### 2.3. Protein-Coding Genes and Codon Usage

The total length of 13 PCGs of *Polypylis* sp. TS-2018 is 10,606 bp, comprising 77.2% of the whole mitogenome. The lengths of individual PCGs range from 114 bp of *atp8* to 1656 bp of *nad5* (Table 1). The A + T content of PCGs is 74.6%, which exhibits strong AT bias. All PCGs initiate with the typical start codon ATN except *nad4* and *nad5*, which starts with TTG (Table 1). The TTG start codon is an alternative to the more commonly used start codons, and it is also present in *nad5* of other ramshorn snails, such as *Biomphalaria choanomphala* [13] and *Gyraulus* sp. (GenBank accession number MW357851). Most PCGs of *Polypylis* sp. TS-2018 utilizes TAA or TAG as a stop codon, but *atp6*, *cox3*, *nad1*, and *nad2* use an incomplete stop codon T (Table 1). The presence of incomplete stop codons TA or T is a common feature of the mitochondrial genes in most gastropods [33]. The incomplete stop codon TA has recently been found in *cob* of *Bulinus truncatus* [14]. The incomplete stop codons are thought to be completed as TAA via post-transcriptional polyadenylation [34].

The 13 PCGs of *Polypylis* sp. TS-2018 comprises a total of 3525 codons. The RSCU analysis reveals Leu, Phe, and Ile are the three most frequently used amino acids, and the most frequently used codons are UUA, AGA, and UCU (Figure 2). In contrast, the codons CGC, CUC, and CUG are rarely used. The most frequently used codons are composed of A or T, and the rarely used codons are composed of C or G. This indicates that the RSCU values are positively correlated with the AT bias of PCGs. The phenomenon is also commonly observed in many species. For example, Zhou et al. [3] found that codons ending in A or T in the freshwater mussel *Novaculina chinensis* were preferred according to RSCU values. DNA with an AT bias has a lower stability and a higher mutation rate [35,36].

### 2.4. Transfer and Ribosomal RNA Genes

The total length of the 22 tRNAs is 1354 bp, with each tRNA ranging from 55 to 70 bp. Most tRNAs can fold into the typical cloverleaf structure except for *trnS1* and *trnS2*, which lack the dihydrouridine (DHU) arm (Figure 3). This phenomenon seems to be a common feature in gastropod mitogenomes [33]. Some base pairs are not the classic bonds of A-U and C-G in the tRNA secondary structure. We observe 22 G-U wobble pairs, 13 U-U mismatches, 3 U-C mismatches, and 1 A-C mismatch in total in 15 tRNAs. Mismatched and wobble pairs are commonly found in invertebrate tRNAs, such as insects [37], bivalves [3], and gastropods [5]. These mismatched pairs are thought to be corrected through post-transcriptional RNA editing [38,39].

The lengths of *rrnL* and *rrnS* of *Polypylis* sp. TS-2018 are 1006 bp and 709 bp, respectively. *rrnL* is located between the *trnV* and *trnL1*, while *rrnS* is located between *trnE* and *trnM*. The A + T content of *rrnL* is 79.6%, while the A + T content of *rrnS* is 75.7%. *rrnL* shows negative AT skews and positive GC skews, whereas *rrnS* shows both negative AT skews and GC skews. *rrnL* has been extensively used as a molecular marker for phylogenetic research of ramshorn snails. Attwood et al. [11], Saito et al. [7], and Martin et al. [40] investigated the distribution and population diversity of ramshorn snails via phylogenetic analysis using *rrnL*. In addition, as *rrnL* is conserved in bacteria and archaea, it has been widely used as a potential molecular marker for taxon identification of bacteria and archaea [28]. For example, Clerissi et al. [41] used *rrnL* to detect microeukaryotes communities associated with ramshorn snails of *Biomphalaria glabrata* and found that the microeukaryotes of Amoebozoa, Opisthokonta, and Alveolata were dominantly associated with *Biomphalaria glabrata*.

### 2.5. Selective Pressure Analysis among Planorbidae

The ratio of nonsynonymous (Ka) and synonymous (Ks) substitution rates (Ka/Ks) is a measure used in molecular evolution to assess the selective pressure acting on PCGs [42]. The Ka/Ks > 1 indicates positive selection, Ka/Ks = 1 indicates neutral evolution, and Ka/Ks < 1 indicates negative and purifying selection [42]. In our study, we have calculated the Ka/Ks of 13 PCGs from each genus of *Planorbidae* to investigate the evolutionary traits of selective pressure. The Ka/Ks values of most PCGs in different genera are <1, which indicates a strong purifying and negative selection in the ramshorn snails (Figure 4) [42]. However, the Ka/Ks ratios of *atp8* in *Anisus*, *Biomphalaria*, and *Gyraulus* are >1 (Figure 4), which indicates the existence of positive selection [42]. The overall average Ka/Ks values of each PCG show that the highest average Ka/Ks value is found for *atp8* (0.802), followed by *nad6* (0.562), *nad2* (0.552), *nad4L* (0.523), *nad5* (0.422), *nad4* (0.407), *nad3* (0.372), *atp6* (0.328), and *nad1* (0.307). The genes *cob*, *cox2*, and *cox3* show lower values of 0.147, 0.145, and 0.159, respectively. The *cox1* gene shows the lowest average Ka/Ks value (0.068) and has little variation among genera (Figure 4), suggesting that *cox1* is very conserved and undergoes strong purifying selection. Saccone et al. [43] found that *atp8* was the most variable, followed by *atp6* and NADH dehydrogenase genes, and *cob*, *cox1*, *cox2*, and *cox3* were more conserved mitochondrial genes in most metazoans. Here, we observed similar evolutionary patterns in ramshorn snails. In this case, we suggest *cob*, *cox1*, *cox2*, and *cox3* may be evaluated as potential molecular markers for species identification in ramshorn snails. The NADH dehydrogenase genes with higher genetic variations can be candidates for intra-specific molecular studies for the ramshorn snails, especially *nad2*, *nad4*, *nad5*, and *nad6* due to their long sizes.

### 2.6. Mitochondrial Gene Arrangement among Planorbidae

Gene rearrangements of gastropod mitogenomes are usually substantially different between major lineages, but species in the same major lineage are relatively conserved [33]. There are four main gene rearrangement events in the mitogenomes of animals, i.e., gene reversals, transpositions, reverse transpositions, and tandem duplication random loss [44]. By comparing the gene orders of 15 *Planorbidae* species, we observe that transpositions and reverse transpositions occurred on 10 tRNAs (*trnC*, *trnD*, *trnE*, *trnL2*, *trnM*, *trnN*, *trnQ*, *trnR*, *trnS2*, *trnW*), and *rrnS*, which formed six gene arrangement types (Figure 5). Among these types, type B is the most commonly observed gene arrangement in gastropod species such as *Albinaria coerulea* [45], *Meghimatium bilineatum* [46], and *Onchidella celtica* [33].

As shown in type A, *Bulinus globosus*, *Bulinus nasutus*, and *Bulinus ugandae* have an identical gene arrangement in their mitogenomes. *Bulinus truncatus* shares an identical gene arrangement with the six species of *Biomphalaria* and two species of *Planorbella* as type B, which is the most common gene arrangement in ramshorn snails. Compared to type A, type B features a transposition of *trnD* to the gene junction between *cob* and *trnC*. We observe that type C exists only in *Biomphalaria tenagophila*, which exhibits reverse transposition of seven tRNAs (*trnE*, *trnL2*, *trnM*, *trnN*, *trnQ*, *trnR*, and *trnS2*) based on type B. *Anisus vortex*, *Polypylis* sp. TS-2018 and *Gyraulus* sp. show a similar gene arrangement with the transposition of *trnW* and *trnC* to the gene junction between *nad4L* and *cob* (type D, type E, and type F, respectively). Moreover, the reverse transposition of *rrnS* exists both in *Polypylis* sp. TS-2018 and *Gyraulus* sp., while the reverse transposition of *trnQ* exists only in *Gyraulus* sp. The mitochondrial gene arrangement observed in *Biomphalaria tenagophila*, *Anisus vortex*, *Polypylis* sp. TS-2018, and *Gyraulus* sp. is very distinctive compared to the other mollusk mitogenomes sequenced so far. Due to the limited available mitogenomes of *Planorbidae*, it is necessary to further sequence additional *Planorbidae* mitogenomes to confirm the uniqueness of the gene arrangements in these species.

### 2.7. Phylogenetic Analysis

We have reconstructed the phylogenetic relationships of 15 ramshorn snails from six genera (*Anisus*, *Biomphalaria*, *Bulinus*, *Gyraulus*, *Planorbella*, and *Polypylis*) by Bayesian inference (BI) and maximum likelihood (ML) methods using the PCGs and rRNAs by dividing them into four datasets (P123, P123R, P12, and P12R). The best partitioning scheme and optimal substitution models for each dataset are shown in Appendix A. *Radix auricularia* (GenBank accession number KP098540) was employed as an outgroup taxon.

Both BI and ML trees recover identical topologies with high posterior possibilities and bootstrap values (Figure 6). Each genus forms a monophyletic clade on the trees. The *Polypylis* sp. TS-2018 (Segmentinini) sequenced in this study is sister to the clade of *Anisus vortex* + *Gyraulus* sp. (Planorbini), and then clusters with the clade of the genera *Planorbella* + *Biomphalaria* (Helisomatini) (Figure 6). The *Bulinus* species occupy a basal position (Figure 6). Molecular phylogenetic analysis of *Planorbidae* using *cox1* and *18S rDNA* had a similar result [10], as various mitochondrial and nuclear markers were analyzed [18,19,20,50]. *Bulinus* was considered to belong to the subfamily Bulininae by Hubendick [17], Morgen et al. [19], and Jørgenson et al. [18] based on anatomical characteristics of the male reproductive system and molecular data. But Albrecht et al. [10] upgraded Bulinidae to family level, and this change was adopted by Bouchet and Rocroi [20]. Nevertheless, our data support the genus *Bulinus* as an ancient divergent taxon, whatever its taxonomic rank.

## 3. Materials and Methods

### 3.1. Samples Collection and DNA Extraction

The specimens used in this study were collected from Shaoxing, Zhejiang, China (30°8′3″ N, 120°35′57″ E). The samples were stored in 100% ethanol at −20 °C and deposited at the Zhejiang Provincial Key Laboratory of Biometrology and Inspection & Quarantine, China Jiliang University, Hangzhou, China. We extracted the genomic DNA from the soft body of an individual using the TIANamp Genomic DNA kit (TIANGEN, Beijing, China) according to the manufacturer’s instructions. The purified genomic DNA was characterized using the Nanodrop 2000 spectrophotometer (Thermo Fish Scientific, Wilmington, DE, USA).

### 3.2. Species Identification

We identified the species based on DNA barcoding analysis. A gene fragment of mitochondrial *cox1* was amplified using the primers LCO1490 (5′-GGTCAACAAATCATAAAGATATGG-3′) and HCO2198 (5′-TAAACTTCAGGGTGACCAAAAAATCA-3′) [54]. The PCR protocol was initial denaturation at 94 °C for 10 min, followed by 34 cycles of denaturation at 94 °C for 30 s, primer annealing at 50 °C for 30 s, and extension at 72 °C for 1 min, and the final extension at 72 °C for 10 min. We purified the PCR products using the TIANquick Midi Purification Kit (TIANGEN, Beijing, China) according to the manufacturer’s instructions and then sequenced them by LC-Biotechnologies, Co., Ltd. (Hangzhou, China).

After checking the reliability of the nucleotides of the sequences, we submitted the confirmed *cox1* sequence for nucleotide BLAST (BLASTn) in NCBI (http://www.ncbi.nlm.nih.gov/, accessed on 15 October 2023) [27] and the BOLD species identification system (https://www.boldsystems.org, accessed on 15 October 2023) [55] and confirmed the species identity as *Polypylis* sp. TS-2018.

### 3.3. Mitogenome Sequencing and Assembly

We performed high-throughput sequencing of *Polypylis* sp. TS-2018 using the Illumina MiSeq platform created by LC-Biotechnologies, Co., Ltd. (Hangzhou, China) and obtained 5.2G bases with an average read length of 144 bp. The sequencing data have been deposited in NCBI under accession number SRR26384977.

We assembled the circular mitogenomes using Geneious Prime 2023.0.1 (https://www.geneious.com, accessed on 21 March 2023) [56] using the *cox1* sequence as the initial reference. The contigs were extended using the assembly parameters of the minimum overlap of 30 bp and the minimum overlap similarity of 100% until the whole circular chromosome was obtained.

### 3.4. Mitogenome Annotation

We used the MITOS Web Server [57] to predict the orientation, size, and position of each mitochondrial gene under the genetic code “05-invertebrate”. The start and stop codons of protein-coding genes (PCGs) were manually checked by searching for open reading frameworks (ORFs) with the Find ORFs tool implemented by Geneious [56]. We manually checked the tRNAs based on secondary structures and identified the rRNAs via the boundary of flanking genes. Finally, we confirmed all genes by aligning with homologous genes of *Planorbidae* species available in GenBank. We drew the circular graph of the mitogenome using the online CGView tools [58]. We calculated nucleotide composition for the complete mitogenome, PCGs, tRNAs, and rRNAs in Geneious [56], and measured the skewness using the following formulas: AT skew = (A − T)/(A + T) and GC skew = (G − C)/(G + C) [59]. The relative synonymous codon usage (RSCU) and amino acid frequencies of PCGs were calculated to understand the genetic codon bias using CodonW (https://codonw.sourceforge.net/, accessed on 17 November 2023) [60]. All stop codons were removed from the calculation to prevent bias brought on by incomplete stop codons.

### 3.5. Comparative Mitogenomic Analysis

We downloaded all 32 published complete mitogenome sequences of 14 *Planorbidae* species from GenBank for comparative analysis. Our preliminary analyses revealed mitogenomes for the same species with a sequence similarity >99.8%. Thus, we selected one representative sequence from each species for further analysis.

In total, we included 14 mitogenome sequences from five genera of *Planorbidae*, namely one species of *Anisus*, six species of *Biomphalaria*, four species of *Bulinus*, one species of *Gyraulus*, and two species of *Planorbella* (Table 2). Because the mitogenome sequences of the *Anisus vortex* lacked annotation, we annotated them according to the previously described methods. We calculated the nonsynonymous (Ka) and synonymous (Ks) substitution rates among the 13 PCGs of the *Planorbidae* species using DnaSP 6.0 [61] and compared the gene rearrangement and the ratio of Ka/Ks among the species. *Radix auricularia* (GenBank accession number KP098540) from the family Lymnaeidae was used as a reference sequence in the substitution calculation.

### 3.6. Phylogenetic Tree Reconstruction

We reconstructed phylogenetic relationships using 14 species from five genera of *Planorbidae* species from GeneBank and the new sequence from this study (Table 2). We employed *Radix auricularia* (GenBank accession number KP098540) from the family Lymnaeidae as an outgroup. We aligned the amino acid sequences of the 13 PCGs and the nucleotide sequences of two rRNAs individually using ClustalW implemented in MEGA X [62]. We concatenated the alignments into four datasets: (1) the P123 matrix, 10,476 residues of the three codon positions of 13 PCGs; (2) the P123R matrix, 12,037 residues of the three codon positions of 13 PCGs plus two rRNAs; (3) the P12 matrix, 6994 residues of 13 PCGs excluding the third codon sites; (4) the P12R matrix, 8555 residues of 13 PCGs excluding the third codon sites, plus two rRNAs.

We reconstructed phylogenetic trees using Bayesian inference (BI) using MrBayes v3.2.7a software [63] and maximum likelihood (ML) methods using the IQtree web server (http://iqtree.cibiv.univie.ac.at/, accessed on 24 November 2023) [64]. The best partition schemes and optimal substitution models for each of the four datasets were identified using PartitionFinder v.2.1.1 [65] with the corrected Akaike Information Criterion (AICc). The “greedy” scheme was used along with branch lengths estimated as “linked”. For the BI trees, two runs of four Markov Chain Monte Carlo (MCMC) chains were run simultaneously for 10,000,000 generations, with a relative burn-in of 25% using the best-fit partitioning scheme. The MCMC process was terminated when the average standard deviation of split frequencies fell below 0.01. For the ML trees, we analyzed 1000 bootstrap replicates in ultrafast likelihood bootstrap. The phylogenetic trees were visualized using FigTree v1.4.4 (http://tree.bio.ed.ac.uk/software/figtree/, accessed on 24 November 2023) [66].

## 4. Conclusions

The ramshorn snails from *Planorbidae* have an extensive species diversity of great ecological and medical importance. However, there are only 14 species with complete mitogenomes, though several that are close to completion were published in GenBank prior to the present study. In this study, we assembled the high-quality whole mitogenome of *Polypylis* sp. TS-2018, which represents the first mitogenome from the genus *Polypylis*. This newly sequenced mitogenome shares similar mitogenome features with other reported *Planorbidae* mitogenomes. Evolutionary analysis reveals that most PCGs in different genera indicate a purifying and negative selection, while *atp8* in *Anisus*, *Biomphalaria*, and *Gyraulus* indicates the existence of positive selection. We have observed six types of gene arrangements in 15 *Planorbidae* species, which are the result of two rearrangement events: inverted transposition and transposition. The phylogenetic analysis indicates that the species of *Bulinus* form the basal clade. *Polypylis* sp. TS-2018 has a close relationship with *Anisus vortex* + *Gyraulus* sp., while *Biomphalaria* + *Planorbella* clade is placed as a sister clade to them. Our results provide important information for further studying species identification, population genetics, and phylogenetic relationships not only of the genus *Polypylis* but also of other ramshorn snails.

## Figures and Tables

**Figure 1 ijms-25-02279-f001:**
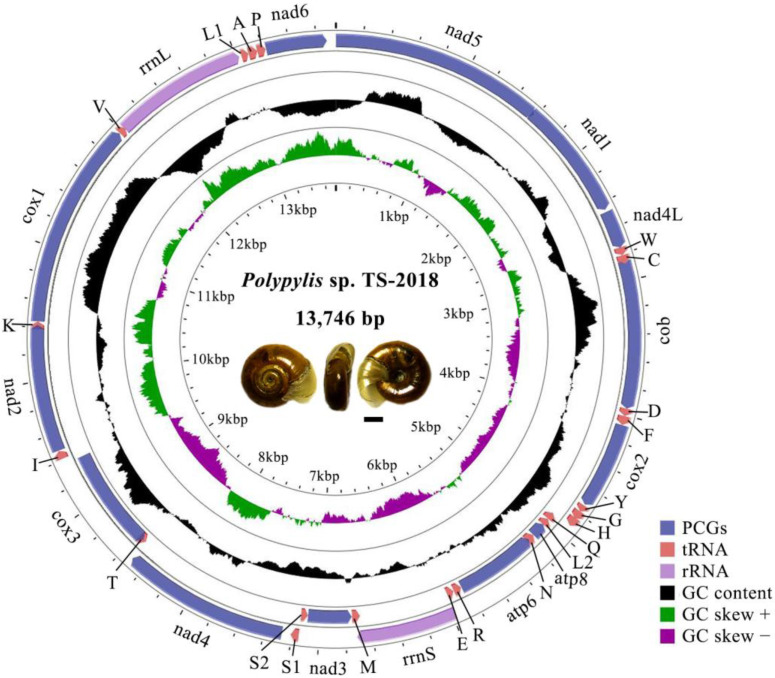
Mitogenome map of *Polypylis* sp. TS-2018. Arrows indicate the orientation of gene transcription. The protein-coding genes (PCGs) are shown as blue arrows, transfer RNA (tRNA) genes as pink arrows, and ribosomal RNA (rRNA) genes as purple arrows. The GC content is plotted using a black sliding window as the deviation from the average GC content of the entire sequence. The GC skew is plotted using a colored sliding window (green and orchid color) as the deviation from the average GC skew of the whole sequence. Abbreviations of gene names are: *atp6* and *atp8* for ATP synthase subunits 6 and 8, *cox1*–*3* for cytochrome *c* oxidase subunits I-III, *cob* for cytochrome b, *nad1*–*6* and *nad4L* for NADH dehydrogenase subunits 1–6 and 4L, *rrnL*, and *rrnS* for large and small rRNA subunits. tRNA genes are indicated with one-letter corresponding amino acids; the two tRNA genes for leucine and serine have different anticodons. The shell morphology of *Polypylis* sp. TS-2018 is an original image by the authors. Scale bar = 0.5 mm.

**Figure 2 ijms-25-02279-f002:**
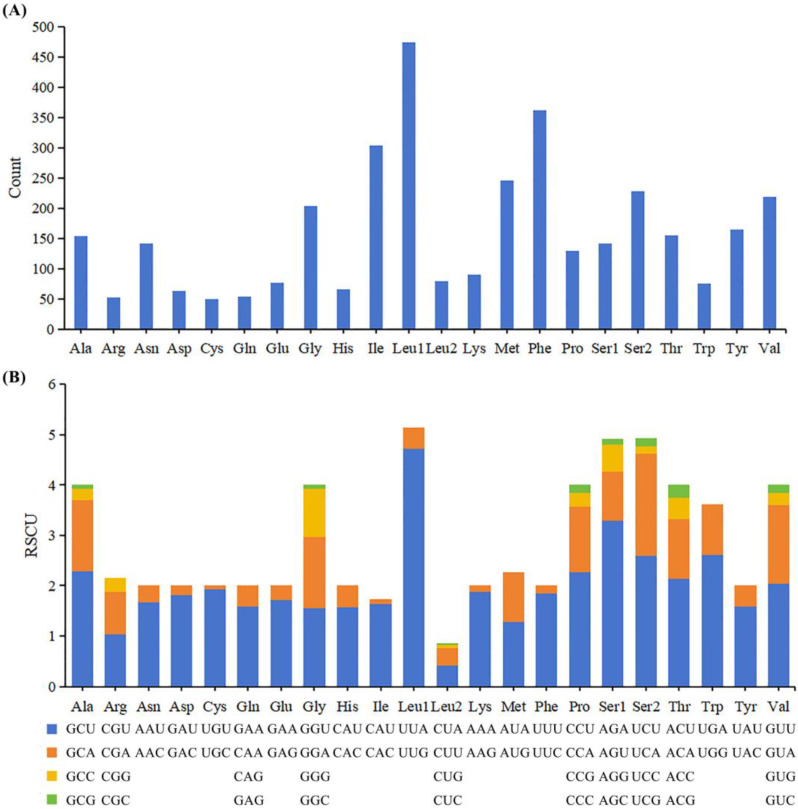
Amino acid composition (**A**) and relative synonymous codon usage (RSCU) (**B**) in the 13 protein-coding genes of *Polypylis* sp. TS-2018. The *y*-axis shows RSCU values, while the *x*-axis shows families of synonymous codons and their corresponding amino acids.

**Figure 3 ijms-25-02279-f003:**
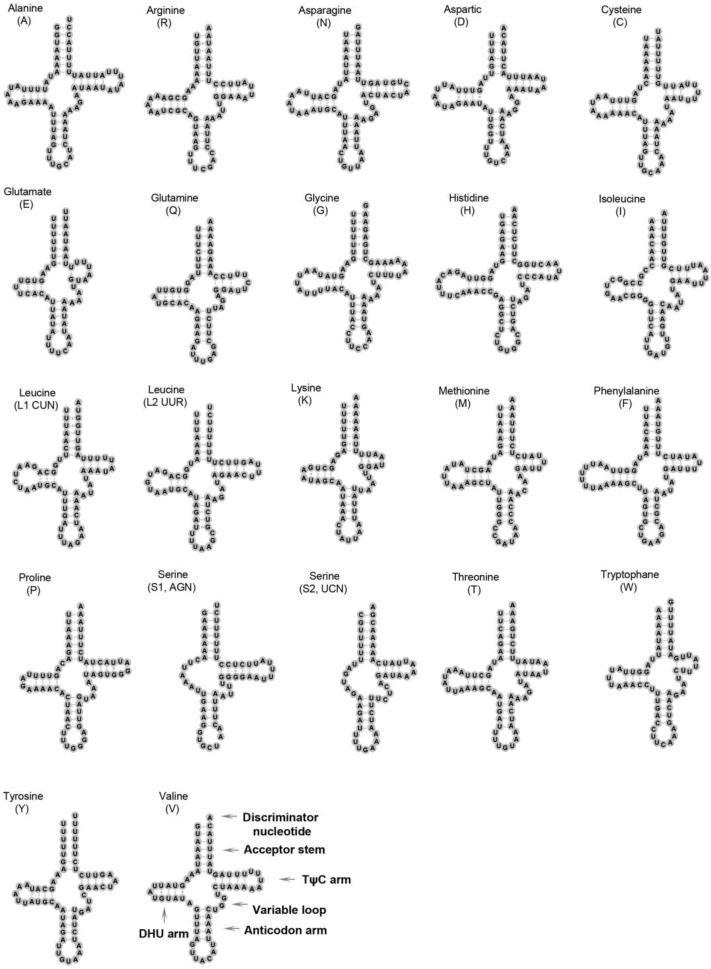
The secondary structures of 22 tRNA genes inferred for the mitogenome of *Polypylis* sp. TS-2018. The tRNAs are labeled with their corresponding amino acids. Dashes (–) indicate Watson–Crick bonds, and dots (·) indicate G-U wobble nucleotide bonds.

**Figure 4 ijms-25-02279-f004:**
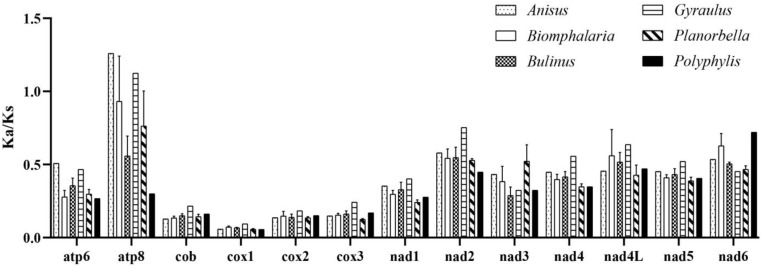
Evolutionary rates (Ka/Ks) of 13 protein-coding genes of *Planorbidae* species.

**Figure 5 ijms-25-02279-f005:**
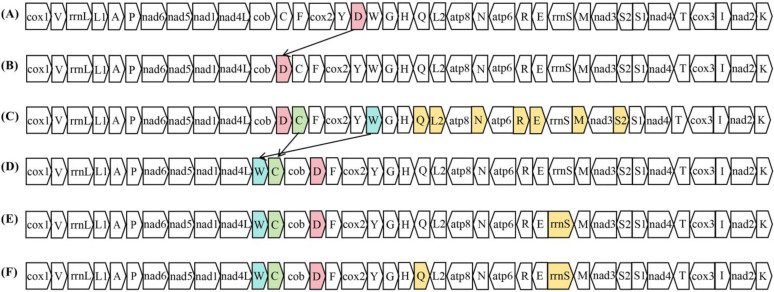
Mitochondrial gene arrangement of *Planorbidae* species. The gene order of (**A**) *Bulinus globosus* [14], *Bulinus nasutus* [14], and *Bulinus ugandae* [14]; (**B**) *Biomphalaria choanomphala* [13], *Biomphalaria glabrata* [47], *Biomphalaria pfeifferi* [13], *Biomphalaria straminea* [13], *Biomphalaria sudanica* [13], *Bulinus truncatus* [13], *Planorbella duryi* [48] and *Planorbella pilsbryi* [30]; (**C**) *Biomphalaria tenagophila* [49]; (**D**) *Anisus vortex* [29]; (**E**) *Polypylis* sp. TS-2018 (GenBank accession number OR684570, reported in this study); (**F**) *Gyraulus* sp. (GenBank accession number MW357851). Colored boxes indicate the events of gene rearrangement.

**Figure 6 ijms-25-02279-f006:**
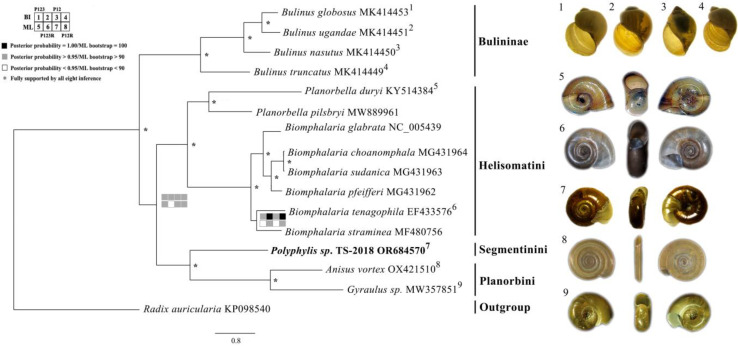
The phylogenetic relationships of *Planorbidae* species are inferred from ML and BI analysis based on the four matrices (P123R, P123, P12R, and P12). *Radix auricularia* (GenBank accession number KP098540) was employed as an outgroup taxon. * indicates full support of all eight inferences (posterior probabilities = 1.00 in four BI trees and bootstrap = 100 in four ML trees). Shells of representatives of the major clades correspond to species in the phylogeny according to the small numbers (numbers 1–9). The shell morphology of *Anisus vortex*, four *Bulinus* species, *Biomphalaria tenagophila*, and *Gyraulus* sp. is taken by Skipp and Ablett [29], Zhang et al. [14], Ovando and Marchi [51], Sitnikova et al. [52], respectively. The shell morphology of *Planorbella duryi* is downloaded from the web of Australian Freshwater Molluscs (https://keys.lucidcentral.org/keys/v3/freshwater_molluscs/, accessed on 29 December 2023) [53]. The shell morphology of *Polypylis* sp. TS-2018 is an original image by the authors.

**Table 1 ijms-25-02279-t001:** Characteristics of *Polypylis* sp. TS-2018 mitogenome.

Gene ^1^	Location	Size (bp)	INC ^2^ (bp)	AT%	AT Skew	GC Skew	Start Codon	Stop Codon	Anti-Codon
*nad6*	1–450	450	0	82.0%	−0.247	0.210	ATA	TAA	
*nad5*	443–2098	1656	−8	76.3%	−0.172	0.087	TTG	TAG	
*nad1*	2079–2973	895	−20	73.5%	−0.201	0.139	ATG	T	
*nad4L*	2991–3272	282	17	81.2%	−0.170	0.170	ATA	TAA	
*W*	3271–3330	60	−2	76.7%	−0.087	0.000			TGA
*C*	3331–3394	64	0	87.5%	0.036	0.000			TGC
*cob*	3410–4468	1059	15	71.9%	−0.214	0.013	ATA	TAG	
*D*	4464–4527	64	−1	80.4%	0.022	0.455			GAC
*F*	4524–4591	68	0	76.5%	−0.077	0.125			TTC
*cox2*	4593–5237	645	1	72.6%	−0.124	0.006	ATT	TAA	
*Y*	5241–5303	63	8	71.7%	0.000	0.067			TAC
*G*	5298–5363	66	−1	78.8%	0.000	0.000			GGA
*H*	5357–5426	70	−7	55.7%	0.026	0.032			CAC
* Q *	5431–5491	61	4	67.2%	−0.073	−0.200			CAA
* L2 *	5487–5553	67	0	71.2%	−0.048	−0.294			TTA
* atp8 *	5549–5662	114	−2	75.4%	−0.256	0.214	ATC	TAA	
* N *	5664–5732	69	1	79.7%	0.091	−0.143			AAC
* atp6 *	5733–6366	634	0	74.9%	−0.162	0.057	ATT	T	
* R *	6370–6433	64	3	71.9%	0.130	0.000			CGA
* E *	6437–6492	56	3	83.9%	−0.191	−0.111			GAA
*rrnS*	6493–7201	709	0	75.7%	−0.017	−0.047			
* M *	7202–7264	63	0	71.4%	0.111	0.222			ATG
* nad3 *	7267–7611	345	2	80.3%	−0.141	−0.088	ATT	TAA	
* S2 *	7612–7666	55	0	76.4%	0.048	−0.077			TCA
*S1*	7666–7725	60	−1	73.3%	−0.136	0.000			AGC
*nad4*	7725–9029	1305	−1	76.9%	−0.163	0.136	TTG	TAA	
* T *	9041–9105	65	11	81.5%	0.132	−0.167			ACA
* cox3 *	9106–9883	778	0	70.4%	−0.131	−0.087	ATG	T	
*I*	9927–9990	64	43	62.5%	−0.050	0.167			ATC
*nad2*	9992–10901	910	1	77.6%	−0.255	0.176	ATT	T	
*K*	10902–10959	58	0	82.8%	−0.083	0.400			AAA
*cox1*	10959–12491	1533	−1	69.8%	−0.264	0.106	ATC	TAA	
*V*	12494–12548	55	2	81.8%	−0.156	0.200			GTA
*rrnL*	12549–13554	1006	0	79.6%	−0.049	0.172			
*L1*	13555–13617	63	16	78.2%	−0.023	0.167			CTA
*A*	13614–13680	67	0	85.1%	0.018	0.200			GCA
*P*	13681–13745	65	0	72.3%	0.064	0.333			CCA

^1^ Genes underlined are on the minor strand; genes not underlined are on the major strand; tRNA genes are indicated with one-letter corresponding amino acids. ^2^ INC: intergenic nucleotides; positive values indicate gaps and negative values indicate overlapped nucleotides between adjacent genes.

**Table 2 ijms-25-02279-t002:** The *Planorbidae* species used in this study.

Genus	Species	GenBank No.	Length (bp)	Reference
*Anisus*	*Anisus vortex*	OX421510	13,566	[29]
*Biomphalaria*	*Biomphalaria choanomphala*	MG431964	13,672	[13]
	*Biomphalaria glabrata*	NC_005439	13,670	[47]
	*Biomphalaria pfeifferi*	MG431962	13,624	[13]
	*Biomphalaria straminea*	MF480756	13,650	[13]
	*Biomphalaria sudanica*	MG431963	13,671	[13]
	*Biomphalaria tenagophila*	EF433576	13,722	[49]
*Bulinus*	*Bulinus globosus*	MK414453	13,747	[14]
	*Bulinus nasutus*	MK414450	13,690	[14]
	*Bulinus truncatus*	MK414449	13,767	[14]
	*Bulinus ugandae*	MK414451	13,715	[14]
*Gyraulus*	*Gyraulus* sp.	MW357851	13,650	unpublished
*Planorbella*	*Planorbella duryi*	KY514384	14,217	[48]
	*Planorbella pilsbryi*	MW889961	13,720	[30]

## Data Availability

The genome sequence data that support the findings of this study are openly available in GenBank of NCBI at https://www.ncbi.nlm.nih.gov, accessed on 13 November 2023 under accession no. OR684570. The associated BioProject, SRA, and Bio-Sample numbers are PRJNA1028109, SRR26384977, and SAMN37811158, respectively.

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
