# Peer review of "Comparative Mitogenome Analyses of Fifteen Ramshorn Snails and Insights into the Phylogeny of Planorbidae (Gastropoda: Hygrophila)"

_ijms, 2024, doi:10.3390/ijms25042279_

Round 1
Reviewer 1 Report
Comments and Suggestions for Authors
The work is thorough and sound.
- critically analyze the manuscript
The manuscript does not earn much criticism, and what little needs correction is addressed in the attached PDF file, which i also submitted on-line. It is a phylogenetic analysis based on new mitogenomes and it is done sufficiently competently that criticism might be gratuitous. I had anticipated that International Journal of Molecular Sciences would be pleased to publish such work.
All of the comments that the authors need are included in the PDF file that i submitted.
As outlined in the abstract, the mitogenomes and phylogenetic analysis are new, and add new genera to what is still a very young field within the molecular sciences. So there's not much to compare. There is no specific improvement regarding the methodology.
The conclusions are consistent; the main questions are few, simple, and comprehensively addressed.
The figures and tables are collectively an excellent example to many authors of what high quality is possible. They are also not unusual for this field of publication, especially in an MDPI journal.

Please consider the language suggestions on the attached copy.
Author Response
We appreciate the reviewer for the positive comments on our manuscript. We have revised our manuscript according to the reviewer’s suggestion. The correction points are attached in PDF file.

Reviewer 2 Report
Comments and Suggestions for Authors
Dear Editor,
The authors of manuscript sequenced 15 full mitogenomes of Ramshorn snails and performed phylogenetic analysis of Planorbidae. I have some suggestions to improve quality of the paper.
Line 17. How authors compared available mitogenomes of ramshorn snails?
Line 37-38. species identification – snail species identification.
Line 58. high economic – and health..
Line 86. Add the following link https://www.ncbi.nlm.nih.gov/.
At the end of Introduction section, the authors should highlight what provoked the present study.
Line 94. I suggest moving all paragraph regarding cox1 gene structure and phylogeny after 2.2. Mitogenome structure and organization.
Line 98. in Saito et al. – replace with by Saito et al.
Line 111. Why the authors deposited only one from 15 mitogenomes in GenBank? Are all of them are identical?
Species name of legend of Figure 5 should be in italic.
In Figure 6 authors should pointed out the Radix auricularia was chosen as outgroup taxon.
Line 293. Please, replace characterized with spectrophotometrically evaluated.
Line 300. Annealing – primer annealing.
Line 361. IQtree web server – pleased add a following link http://iqtree.cibiv.univie.ac.at/.
Comments on the Quality of English Language
Minor editing of English language required.
Author Response
We thank the comments from the reviewer. Instead of sequencing the mitogenomes of 15 ramshorn snails, we sequenced one complete mitogenome of a ramshorn snail, Polypylis sp. TS-2018. The other 14 mitogenomes of Planorbidae used in this study were retrieved from GenBank. We have shown the sequence information in Table 2. We compared the 15 mitogenomes of ramshorn snails and used them for phylogenetic analysis. We have revised our manuscript following the reviewer’s suggestions. Please see the attached PDF file.

Reviewer 3 Report
Comments and Suggestions for Authors
Dear authors,
well done for this work, it is very interesting and well presented. I am suggesting some minor corrections within the comments of the attached pdf.
best wishes

the english is good
Author Response
We appreciate the reviewer’s time and efforts on our manuscript. We have carefully reviewed and listed the comments provided in the attached PDF and revised the suggested minor corrections accordingly.
